# Immune Prophylaxis Targeting the Respiratory Syncytial Virus (RSV) G Protein

**DOI:** 10.3390/v15051067

**Published:** 2023-04-27

**Authors:** Harrison C. Bergeron, Jackelyn Murray, Aakash Arora, Ana M. Nuñez Castrejon, Rebecca M. DuBois, Larry J. Anderson, Lawrence M. Kauvar, Ralph A. Tripp

**Affiliations:** 1Department of Infectious Diseases, College of Veterinary Medicine, University of Georgia, Athens, GA 30602, USA; 2Department of Biomolecular Engineering, University of California Santa Cruz, Santa Cruz, CA 95064, USA; amnunezc@ucsc.edu (A.M.N.C.);; 3Division of Pediatric Infectious Disease, Emory University School of Medicine, Atlanta, GA 30322, USA; larry.anderson@emory.edu; 4Trellis Biosciences, Redwood City, CA 94063, USA

**Keywords:** RSV, G protein, F protein, mAb, 3D3, 2D10, palivizumab

## Abstract

The respiratory syncytial virus (RSV) causes significant respiratory disease in young infants and the elderly. Immune prophylaxis in infants is currently limited to palivizumab, an anti-RSV fusion (F) protein monoclonal antibody (mAb). While anti-F protein mAbs neutralize RSV, they are unable to prevent aberrant pathogenic responses provoked by the RSV attachment (G) protein. Recently, the co-crystal structures of two high-affinity anti-G protein mAbs that bind the central conserved domain (CCD) at distinct non-overlapping epitopes were solved. mAbs 3D3 and 2D10 are broadly neutralizing and block G protein CX3C-mediated chemotaxis by binding antigenic sites γ1 and γ2, respectively, which is known to reduce RSV disease. Previous studies have established 3D3 as a potential immunoprophylactic and therapeutic; however, there has been no similar evaluation of 2D10 available. Here, we sought to determine the differences in neutralization and immunity to RSV Line19F infection which recapitulates human RSV infection in mouse models making it useful for therapeutic antibody studies. Prophylactic (24 h prior to infection) or therapeutic (72 h post-infection) treatment of mice with 3D3, 2D10, or palivizumab were compared to isotype control antibody treatment. The results show that 2D10 can neutralize RSV Line19F both prophylactically and therapeutically, and can reduce disease-causing immune responses in a prophylactic but not therapeutic context. In contrast, 3D3 was able to significantly (*p* < 0.05) reduce lung virus titers and IL-13 in a prophylactic and therapeutic regimen suggesting subtle but important differences in immune responses to RSV infection with mAbs that bind distinct epitopes.

## 1. Introduction

RSV is a leading cause of respiratory disease affecting the very young, immunocompromised, and elderly [1,2]. In the United States, RSV can cause approximately 58,000 hospitalizations with 100–500 deaths among children younger than 5 years old, and 177,000 hospitalizations with 14,000 deaths among adults aged 65 years or older (CDC Division of Health Informatics and Surveillance, 2022). Despite this substantial disease burden, there is no safe and approved RSV vaccine. The only specific standard of care available is palivizumab (Synagis^®^), a humanized mouse mAb against the RSV fusion F protein used to reduce severe disease which is recommended for high-risk infants or those having other medical problems including heart or lung diseases [3]. Palivizumab binds an RSV F protein epitope, i.e., residues 262–270 [4]. The F and G proteins are the major surface antigens of RSV [5]. RSV G protein is heavily glycosylated and contains at least 289 amino acids (32–33 kDa) depending on the strain [6,7]. For example, the G protein associated with currently circulating strains subgroup A (ON1) and subgroup B (GA) is up to 319 amino acids [8]. The G protein ectodomain consists of two mucin-like stalks which flank a central conserved domain (CCD) having a CX3C chemokine motif and a heparin-binding domain (HBD) [9]. RSV has two subtypes, A and B, based on the G protein sequence [10].

The CCD in the G protein is highly conserved making it an ideal target for antibodies; however, this region is poorly immunogenic [11,12,13]. CCD mutations may enhance immunogenicity in mice, and the CX3C chemokine motif can modify immunity [11,14,15,16]. The CX3C chemokine motif mimics the activities of fractalkine (FKN, CX3CL1) [14] which is used by RSV for attachment to CX3CR1 expressed by respiratory epithelial cells and some immune cells [17,18,19,20]. Importantly, RSV modifies immunity in part through the actions mediated by the G protein by inducing aberrant T-cell trafficking, biasing a pathogenic Th2-type response, antagonizing the antiviral type I and III interferons (IFN) response, and altering proinflammatory responses [21,22,23,24]. The G protein also acts as an antibody decoy because a soluble form of G protein (sG) is released following infection as the G protein gene contains an alternative translation site (Met48) which results in an unstable TM region and release of sG [11,25]. 

The G protein CX3C motif is associated with viral attachment and immune modulation, thus antibodies that target this motif and block the actions of G protein are of great interest. Several studies have shown that antibodies that block G protein CX3C-CX3CR1 interaction prevent G protein-mediated chemotaxis, reduce lung mucin and Th2-type cytokine responses, neutralize RSV A and B strains, and lessen overall disease severity even in enhanced respiratory disease models [26,27,28,29,30,31,32,33,34,35]. Although these mAbs are mouse-derived, work by our group and others has identified highly potent and disease-protective human anti-G protein mAbs [36,37,38,39]. Two high-affinity anti-G protein mAbs are 3D3 and 2D10 which have been shown to be potent RSV-neutralizing mAbs that prevent G protein CX3C-CX3CR1-mediated chemotaxis [38,40]. Structure-guided binding studies have determined newly described conformational epitopes of these two mAbs [40]. 3D3 binds to site γ1 and 2D10 binds to γ2 which are non-overlapping epitopes on the G protein CCD, and mAb binding blocks CX3C-mediated chemotaxis [40]. 3D3 has been more extensively studied and has been the subject of in vivo studies showing neutralization and protection against RSV A2 in BALB/c mice [36,38]. 2D10 (previously termed AT50) had not been evaluated in vivo but was shown to neutralize RSV in human epithetical airway cells (hAECs) [38,39]. As RSV neutralization by mAbs 3D3 and 2D10 differ by γ1 or γ2 site binding, it is also known that the resulting immune responses to RSV infection differ. We showed different type I and III interferon (IFN) responses in mice and mouse lung epithelial cells treated with mAbs 2D10 or 3D3 where 2D10 treatment improved IFN responses in mice, while 3D3 improved IFN responses in RSV-infected respiratory cells and mice [41]. IFNs are well-characterized correlates of protection [22]. Given the absence of in vivo 2D10 studies, and the known differences in RSV neutralization and IFN responses between mAbs 2D10 and 3D3, we sought to investigate protection against RSV disease by evaluating virus neutralization and various immune responses in prophylactic and therapeutic mAb treatments. 

Palivizumab (MedImmune, Gaithersburg, MD), the only licensed anti-RSV mAb in the United States, is a humanized mAb directed against a neutralizing epitope on the RSV F protein (38). The FDA has recently accepted a Biologics License Application (BLA) for nirsevimab for the prevention of RSV lower respiratory tract disease in newborns and infants entering or during their first RSV season and for children up to 24 months of age who remain vulnerable to severe RSV disease through their second RSV season. Nirsevimab (Beyfortus^®^) is a human recombinant mAb with activity against the RSV F protein [42], and is an investigational new product not yet approved by the FDA as it has not been determined to be safe and effective. Thus, palivizumab is the only existing therapy; however, palivizumab efficacy is limited, preventing ~55% of RSV hospitalizations compared with placebo [3]. Thus, newer mAb countermeasures are needed to protect against RSV disease. 

In this study, we examined palivizumab and two human anti-G protein mAbs that bind distinct epitopes in the RSV G CCD region (γ1 [3D3] and γ2 [2D10]) and compared the findings to isotype control mAbs. Mice were prophylactically (24 h prior to infection) or therapeutically (72 h pi) administered 1 mg/kg of anti-RSV mAbs or isotype control mAb. This dose chosen was based on our previous studies that examined neutralization of the RSV long strain in mice prophylactically treated with palivizumab or 3D3 [38]. Mice were intranasally (i.n.) infected with RSV Line19F which has been shown to better recapitulate RSV disease compared to RSV A2 and is associated with IL-13 mucogenic disease [33,43,44,45]. Prophylactic treatment with 2D10, 3D3, or palivizumab neutralized RSV Line19F, which is important as this showed the in vivo neutralization of the γ2-targeting mAb (i.e., 2D10) for the first time. There were subtle differences in the BAL cell types after prophylactics; however, all prophylactic treatments modified pulmonary leukocyte infiltration compared to isotype control treatment. Notably, only therapeutic treatment with mAbs 2D10 and 3D3 (targeting the G protein) significantly neutralized RSV Line19F compared to the isotype control mAb. Unexpectedly, palivizumab treatment at 72 h pi resulted in substantially higher pulmonary leukocytes (BAL cells) compared to isotype mAb control or anti-G protein mAb treatment. Interestingly, the cytokine responses by BAL cells varied following prophylactic or therapeutic mAb treatments. Specifically, prophylactic treatment with mAbs 2D10, 3D3, or palivizumab significantly (*p* < 0.05) reduced IFNγ and IL-13, while mAb 3D3 treatment modestly reduced IL-4 and 2D10 and palivizumab treatment significantly (*p* < 0.05) reduced IL-4. In contrast, therapeutic delivery of only mAb 3D3 significantly reduced IL-13. These results confirm that prophylactic treatment with palivizumab or γ1 anti-G protein mAb (3D3) neutralizes RSV, and also show that an γ2 anti-G protein mAb (2D10) neutralizes RSV. Importantly, the results show that anti-G protein mAbs significantly (*p* < 0.05) neutralize RSV when administered after 72 h pi, and only the γ1 targeting mAb (i.e., 3D3) is able to significantly (*p* < 0.05) reduce the mucogenic cytokine, IL-13.

## 2. Materials and Methods

### 2.1. Mice

Six-to-eight-week-old specific-pathogen-free female BALB/c mice from Jackson Labs were used in all experiments. Mice were housed in microisolator cages and fed sterilized water and food ad libitum. For RSV infection, mice were anesthetized by intraperitoneal (i.p.) administration of tribromoethanol (Avertin, 2% 2,2,2-tribromoethanol, 180 to 250 mg/kg in tertiary amyl alcohol) and i.n. infected with 10^6^ PFU of RSV A2 Line19F in serum-free minimal essential medium (MEM, HyClone). Additionally, 24 h before infection (prophylaxis) or 72 h pi (treatment), mice were i.p. administered PBS (Gibco, Waltham, MA, USA) or 1 mg/kg mAb 2D10, mAb 3D3, palivizumab, or isotype control (InVivoMab hIgG1, BioXCell, Lebanon, NH, USA). Five mice per group were sacrificed on day 5 pi. Mouse studies were performed according to a protocol approved by the University of Georgia Institutional Animal Care and Use Committee (A2022 04-023-Y1-A0, approval date 19 May 2022).

### 2.2. Viruses

RSV A2 Line19F was propagated in HEp-2 cells (free of HeLa cell contamination) where Hep-2 cells were infected at a multiplicity of infection (MOI) of 0.01 [46]. Infected cells on day 4 pi were removed and lysed by sonication. The cell debris was removed by centrifugation at 500× *g* for 7 min at 4 °C. The viral supernatant was immediately stored in single-use aliquots at −80 °C. Viral titers were determined by plaque assay as described below.

### 2.3. Production of Anti-RSV G Protein mAbs 3D3 and 2D10 

Recombinant mAbs 3D3 and 2D10 were expressed by transient transfection in CHO-S cells and purified from the media by immobilized protein A affinity chromatography as described previously [38,40]. Purified mAbs were dialyzed into PBS and concentrated to 1 mg/mL.

### 2.4. Plaque Assays

Briefly, mouse lungs were aseptically removed day 5 pi and individual lung specimens were homogenized at 4 °C in 1 mL of serum-free DMEM (Hyclone, Logan, UT, USA) using a gentleMACS™ Dissociator (Miltenyi Biotec Inc., San Diego, CA, USA). Samples were centrifuged for 10 min at 200× *g* and the supernatants were transferred to new tubes and used immediately or stored at −80 °C. Plaque assays were performed as previously described [46]. Briefly, 10-fold serial dilutions of the lung homogenates were added to 90% confluent Vero E6 cell (ATCC CRL-1586) monolayers. Following adsorption for 2 h at 37 °C, cell monolayers were overlaid with 2% methylcellulose (Sigma Aldrich, St. Louis, MO, USA) media and incubated at 37 °C for 6 days. The plaques were enumerated by immunostaining with a mixture of mAbs, clones 131-2A and 131-2G, against RSV F and G proteins, respectively, followed by a secondary goat anti-mouse IgG antibody conjugated to alkaline phosphatase (Thermo Fisher Scientific, Grand Island, NY, USA). Plaques were developed with 1-Step NBT/BCIP (Thermo Fisher Scientific) at room temperature for 5–10 min. Plaques were counted using a dissecting microscope. The limit of detection for this assay is 50 PFU/mL.

### 2.5. Bronchoalveolar Leukocytes (BAL) 

Mice were anesthetized with Avertin and then euthanized by exsanguination by severing the left axillary artery. BAL was harvested by lavage of the lungs 3 times using a total of 1 mL sterile PBS (Gibco). BAL fluid (BALF) was centrifuged for 5 min × 1000× *g* at 4 °C to separate BAL cells from the supernatant. BAL cells were resuspended in FACS buffer (PBS + 0.8% bovine serum albumin) and enumerated using Trypan blue and a hemocytometer. 

The BAL cells were immunostained for extracellular markers as described [47]. Briefly, BAL cells were blocked with an Fc blocker (anti-CD16/32) in FACS buffer for 15 min at 4 °C and then stained for 45 min at 4 °C with the appropriate combinations of anti-CD3 (17A2), anti-mouse B220/CD45R (RA3-6B2), anti-CD11b (M1/70), anti-mouse Ly-6G (1A8), anti-mouse CD49b/integrin alpha 2 (DX5), SiglecF (E50-2440), or mouse isotype antibody controls (all from BD Biosciences, Franklin Lakes, NJ, USA) diluted in FACS buffer. Cells were washed in FACS buffer and fixed in 2% paraformaldehyde (PFA, Ted Pella, Redding, CA, USA) for 20 min at room temperature. PFA was decanted and cells were resuspended in FACS buffer prior to analysis. Cells were classified based on the following phenotypes: B220+ cells = SSC^lo^ B220^+^, T cells = SSC^lo^ CD3^+^, NK cells = CD3^−^ DX5^+^, CD11b+ cells (common leukocyte marker for monocytes, macrophages, dendritic cells) = CD11b^+^, PMNs (neutrophils) = Ly6G^+^ CD11b^+^ SiglecF^−^, and eosinophils = Ly6G^−^ CD11b^+^ SiglecF^+^ (Appendix A) [33,48,49]. The distributions and patterns of cell surface markers were determined for at least 10,000 events analyzed on a BD LSRII flow cytometer (BD Biosciences, Mountain View, CA, USA), and data were analyzed using FlowJo software (TreeStar, Ashland, OR, USA).

### 2.6. Cytokine Quantification

IFNγ, IL-4, and IL-13 capture ELISAs were performed according to the manufacturer’s instructions (R&D Systems, Minneapolis, MN, USA). Briefly, the capture antibody was coated onto a high-binding ELISA plate (Corning) overnight. The following day, wells were washed 3× with KPL wash buffer (1× diluted in deionized water) (SeraCare, Milford, MA, USA) and blocked overnight at 4 °C with 1% BSA/KPL buffer. Lung homogenates treated with PMSF Protease Inhibitor (Thermo Fisher Scientific, Grand Island, NY, USA), standards, and controls were added to the plates and incubated overnight at 4 °C. 12 h later the wells were washed 3× with KPL wash buffer and a biotinylated detection antibody was added for 2 h at room temperature (RT). Wells were washed and incubated with streptavidin-HRP in the dark for 20 min at RT. Wells were washed 3× with KPL wash buffer and detected with One-Step TMB (ThermoFisher) and stopped with Stop Solution (ThermoFisher). Plates were read on a BioTek plate reader at OD_450_. A standard curve was used to quantify protein concentrations using standards included in commercial kits. 

### 2.7. Gene Expression by Quantitative Polymerase Chain Reaction (qPCR) 

RNA isolation of lung homogenates was performed using a Qiagen RNA isolation kit as described (Qiagen, Germantown, MD, USA). The total RNA was quantified using NanoDrop (ThermoFisher) as described by the manufacturer. cDNA was synthesized by LunaScript (New England Biolabs, Ipswich, MA, USA) according to the manufacturer. To determine changes in gene expression, qPCR was performed using 2× Ultra-Brilliant III SYBR with low ROX (Agilent, Santa Clara, CA, USA) on MX300 Real-Time PCR instrument (Agilent). RANTES (CCL5) primer sets were procured from IDT PrimeTime™ Pre-Designed Primers. Fold changes in gene expression were determined using ΔΔCt method [50], and normalized to ACTB and compared to untreated, mock-infected mice as previously described [51].

### 2.8. Statistics

Groups were compared using the one-way analysis of variance (ANOVA) with Bonferroni’s Correction. *p* ≤ 0.05 was considered statistically significant. Experiments were performed at least twice independently with representative data presented. All statistical analyses were performed using Prism 9 (GraphPad, San Diego, CA, USA). Data are shown as mean ± SEM.

## 3. Results

### 3.1. Prophylactic or Therapeutic Treatment with Anti-G Protein mAbs Neutralize RSV 

RSV G protein is multifunctional mediating virus attachment to host cells, typically by G protein CX3C-CX3CR1 binding to ciliated respiratory epithelial cells [17,52,53], and by modifying immunity and disease pathogenesis [15,23,24]. Importantly, anti-G protein antibodies targeting the G protein CCD are protective when administered prophylactically or therapeutically in animal models [27,35,54]. Here, we examine anti-G protein mAb 2D10 for the first time in vivo and confirm mAbs 3D3 and palivizumab as neutralizing and disease protective [27,35,36,37,38,54]. The two broadly neutralizing human anti-G protein mAbs bind the RSV G protein CCD with high affinity (low picomolar-dissociation constant) at different, non-overlapping epitopes (Figure 1) [40]. 3D3 binds the γ1 site and 2D10 binds the γ2 site [40].

Previous studies have shown that mAbs 3D3 and palivizumab neutralize RSV A2 in mice [36,37,38]. Consistent with these studies, Figure 2A demonstrates that prophylactic treatment with mAb 3D3 or palivizumab neutralizes RSV Line19F leading to decreased lung virus at peak viral replication (i.e., day 5 pi) compared to isotype control mAb. Additionally, prophylactic treatment with mAb 2D10 significantly (*p* < 0.05) neutralized virus compared to isotype control mAb. While all prophylactic mAb treatment groups significantly (*p* < 0.05) reduced RSV Line19F titers, the greatest reductions were observed with palivizumab prophylaxis which resulted in undetectable RSV titers while mAb 3D3 treatment reduced RSV titers by >2.5 logs and mAb 2D10 by >1.5 logs. When mAbs were delivered 72 h post challenge (Figure 2B), mAb treatment neutralized RSV Line19F to various extents. For example, mAbs 3D3 and 2D10 significantly (*p* < 0.05) neutralized RSV Line19F by approximately 1.5 logs, and palivizumab reduced RSV Line19F titers by approximately 0.5 logs (*p* > 0.05). These data are consistent with several reports showing that anti-G protein mAbs significantly neutralize RSV when delivered therapeutically [32,34,36]. Thus, mAbs targeting the G and/or F proteins are capable of neutralizing RSV to various extents, which is dependent upon the time of mAb delivery after virus infection and the viral antigen targeted by the mAb. 

### 3.2. Anti-G Protein mAbs Reduce Pulmonary Leukocytes

In mice, an intranasal infection with RSV Line19F leads to a substantial influx of pulmonary leukocytes that are able to potentiate immune-mediated pathology [33]. To determine how anti-RSV mAbs affect the pulmonary leukocyte response to infection, mice were treated with 1 mg/kg mAbs 3D3, 2D10, palivizumab or isotype control mAbs and then challenged with RSV Line19F. BAL cells were collected at day 5 pi, or the peak of total BAL cell influx [35], and the cell types were enumerated, immunostained, and analyzed by flow cytometry, as previously described [47]. Consistent with earlier studies that investigated an anti-G protein mouse mAb targeting the CX3C motif (mAb 131-2G) [35], prophylactic treatment with anti-G protein mAb 3D3 significantly (*p* < 0.05) reduced the total BAL cell types and numbers at day 5 pi (Figure 3A), and this correlated with reduced immune-mediated pulmonary disease [55]. While anti-G protein mAb 2D10 and palivizumab appreciably reduced the total BAL cell types and numbers at day 5 pi, this did not reach significance (Figure 3A). Mice treated with anti-G protein mAbs at 72 hpi (Figure 3B), did not have low BAL cells compared to isotype control; however, palivizumab treatment increased the concentration of BAL cells compared to isotype control mAbs (*p* = 0.059). This finding was unexpected as previous studies evaluating higher doses of palivizumab therapy (i.e., 5–15 mg/kg) [35,36] resulted in lower BAL cells, however in those studies higher doses of palivizumab treatment were also associated with significant virus neutralization. Thus, mAb treatment and resulting pulmonary leukocytes may be related to the degree of virus neutralization and/or differences between viral targets of the mAbs. 

BAL cell phenotypes were determined using flow cytometry as described in the Methods section. Previous reports have shown that mice prophylactically or therapeutically treated with anti-G protein mAbs had decreased T cells, CD11b+ cells, NK cells, and B220+ cells following RSV A2 challenge or formalin-inactivated RSV (FI-RSV) immunization [31,34]. For mice prophylactically treated with anti-F or anti-G protein mAbs (Figure 4), all cell types evaluated were decreased compared to isotype control (i.e., B220+ cells (Figure 4A), T cells (Figure 4B), CD11b+ cells (Figure 4C), NK cells (Figure 4D), eosinophils (Figure 4E), and PMNs (Figure 4F)). 3D3 mAb prophylaxis resulted in significant (*p* < 0.05) decreases in the number of T cells, CD11b+ cells, NK cells, and eosinophils. 2D10 mAb prophylaxis reduced CD11b+ and NK cells, and palivizumab prophylaxis reduced T cells and NK cells. Therefore, despite recognizing opposing, non-overlapping epitopes on the G protein CCD, 2D10, and 3D3 are similarly poised to prevent substantial BAL cell influx.

For mice therapeutically treated with anti-G protein mAbs there was no change in BAL cell types compared to isotype control mAbs; however, palivizumab treatment induced considerable increases in B220+ cells (Figure 5A), T cells (Figure 5B) (*p* < 0.05), NK cells (Figure 5D), and eosinophils (*p* = 0.051) (Figure 5E) compared to isotype control. Not all cell types were increased to the same magnitude, indicating palivizumab treatment affects lymphocyte influx in the BAL. For example, CD11b+ cells (Figure 5C) and PMNs (Figure 5F) were not substantially increased regardless of treatment. These findings may affect RSV disease and are consistent with a previous report which demonstrated increased eosinophilia and T cells in anti-F protein-treated mice compared to mAb 3G12-treated mice when mAbs were administered 2 days post-RSV A2 infection [56]. CD11b+ cells were significantly (*p* < 0.05) decreased in mice administered anti-G protein mAb compared to palivizumab treatment, and PMNs and B cells were significantly (*p* < 0.05) decreased in mice treated with anti-G protein mAbs compared to isotype control mAbs [56]. These results suggest anti-G protein mAbs are superior to palivizumab in reducing mediators of immune pathology, i.e., BAL cell influx, and these results are consistent with previous reports [35,56]. 

### 3.3. Antibody Treatment Modifies the Cytokine and Chemokine Response during RSV Infection

RSV infection induces cytokine and chemokine expression in the lung, and treatment with mAbs may modify these responses. Concentrations of IFNγ, IL-4, and IL-13 in the lung on day 5 pi in mice prophylactically or therapeutically treated with anti-RSV mAbs or isotype control mAbs were evaluated by ELISA. IFNγ is a Th1-type cytokine, and IL-4 and IL-13 are Th2-type cytokines. IFNγ is the only member of type II interferon (IFN) and is involved in the activation of macrophages, Th1 cells, and cytotoxic T lymphocytes (CTLs) [57]. IL-4 reduces Th1 cell proliferation, activates IgE class switching, and is generally associated with a more severe RSV disease via Th2-type responses [58]. IL-13 is a mucosal-associated cytokine that is known to influence asthma and is suggested to contribute to Line19F disease in mice [59]. Prophylactic delivery of anti-G protein and anti-F protein mAbs resulted in a significant (*p* < 0.05) decrease in IFNγ and IL-13, while mAb 2D10 and palivizumab significantly (*p* < 0.05), and mAb 3D3 treatment moderately reduced IL-4 (Figure 6A–C). RANTES (regulated on activation, normal T cell expressed and secreted) is an inflammatory chemokine responsible for recruiting leukocytes including T cells and eosinophils and is associated with more severe RSV disease [60]. All prophylactic groups had reduced RANTES transcripts to near mock infected levels as determined by ΔΔCt PCR while isotype-treated mice had a 2.5-fold increase; however, these were not statistically significant (Figure 6D). 

Mice treated with mAbs 2D10, 3D3, or palivizumab at day 3 pi had no significant reductions in IFNγ (Figure 7A) or IL-4 (Figure 7B); however, mAb 3D3 treatment significantly (*p* < 0.05) decreased IL-13 (Figure 7C) compared to isotype control mAb. RANTES transcripts were increased (but not significantly) in palivizumab-treated mice compared to isotype and anti-G protein mAb-treated mice (Figure 7D). An increase in RANTES expression may contribute to the increased BAL cell numbers, i.e., T cells, NK cells, and eosinophils. These data show that anti-RSV mAbs reduce the IFNγ and IL-13 responses when delivered prophylactically, and mAbs 2D10 and palivizumab significantly (*p* < 0.05) reduce IL-4. Only mAb 3D3 was able to significantly (*p* < 0.05) reduce IL-13, a mucogenic cytokine associated with RSV Line19F pathology, when delivered on day 3 pi. 

## 4. Discussion

Anti-F protein mAbs predominate over anti-G protein mAbs in therapeutic development because they were first shown to neutralize RSV, and the F protein has greater conservation, greater immunogenicity, and is required for infection [21]. Palivizumab is a neutralizing mAb that targets site II present on the pre- and post-fusion conformations of the F protein and is currently the only specific prophylactic available to reduce RSV disease in the United States [61]. While palivizumab has been prescribed for use in high-risk infants since the late 1990s, its efficacy and cost analysis have suggested improvements are needed [62,63,64,65]. Specifically, palivizumab reduces hospitalization by ~55% and is not approved for post-exposure therapy or to prevent RSV-mediated asthma [66,67]. Nirsevimab has been evaluated in late-stage clinical trials and designated as a breakthrough therapy by the FDA to prevent RSV in term and pre-term infants; however, it is not investigated as a treatment modality [42,68]. Nirsevimab prevented 74.5% of medically attended RSV-associated LRTI in a recent clinical trial, improving the protection that is provided by palivizumab. 

The G protein contains a CCD region with a CX3C chemokine motif which can function to attach RSV to CX3CR1 and mediate aberrant and potentially pathogenic immune responses [14,17,69,70,71,72]. The CCD region is devoid of glycosylation and more conserved making it an ideal target for neutralizing antibodies [40,70]. Importantly, the CX3C motif on G protein mediates CX3C-CX3CR1 attachment on ciliated respiratory epithelial cells and some immune cells including neonatal B regulatory cells [71,73,74]. RSV has also been shown to infect neuronal cell processes that innervate the mouse lung which is mediated by the RSV G protein [75]. The G protein is a chief mediator of immune dysregulation. For example, the G protein CX3C motif is known to affect lung leukocyte trafficking, compete with FKN for CX3CR1 binding, and bias a mucosal and pathogenic Th2 response [14,15]. sG is secreted as early as 6 h after infection and contributes to immune dysregulation, specifically through modifying macrophage responses and inhibiting antibody function as an antigenic decoy [25]. 

Importantly, the RSV G protein is an IFN antagonist which has implications in RSV immune-mediated disease as IFNs and Th1-type responses have been correlated with protection from diseases [22,76,77]. Recently, we examined types I and III IFNs in mouse lung epithelial (MLE-15) cells and BALB/c mice infected with RSV Line19F and treated with mAbs 3D3, 2D10, palivizumab, or isotype control [41]. Treatment with mAb 3D3 led to improved type I IFN responses in mice and cells, and mAb 2D10 improved type I IFN responses in vivo. For type III IFNs, mAb 3D3 treatment improved the MLE-15 cell response while mAb 2D10 improved the type III IFN responses in mice. IFN modification in MLE-15 cells by mAb 3D3 treatment was linked to the suppressor of cytokine signaling 1 (SOCS1), and as mAbs 3D3 and 2D10 are non-neutralizing in these cells, the IFN improvements were independent of neutralization. These findings support anti-G protein antibodies in improved antiviral protection which has important implications for vaccine and therapeutic design. The improved IFN responses were observed 24 h pi and are consistent with another report showing that anti-G protein mAb (131-2G) targeting G protein CX3C improves type I and III IFN responses [19]. These results support the role of RSV G protein antagonizing protective immunity and demonstrate that mAbs targeting the CCD may improve RSV disease.

As the G protein is important in virus attachment, infection, and immune modulation, it is proposed that anti-G protein mAb therapeutics is key in blocking infection and disease pathogenesis. Notably, several studies have shown that anti-G protein antibodies cross-neutralize RSV A and B strains, reduce indicators of RSV disease such as BAL cell numbers, improve antiviral cytokine and chemokine responses, attenuate pathogenic responses, and balance Th1/Th2 responses, and these immune responses correlate with improved lung pathology following RSV infection [21,30,48,78,79]. Additionally, anti-G protein mAbs have been shown to more effectively treat RSV disease compared to palivizumab, or an anti-F protein mAb in BALB/c mice [35]. We extend these studies by investigating two human anti-G protein mAbs, 3D3 and 2D10. Both 3D3 and 2D10 mAbs were identified as potent neutralizing anti-G protein mAbs which recognize the CCD region of RSV A or B strains and inhibit CX3C-CX3CR1 chemotaxis [40]. The co-crystal structures of 3D3 and 2D10 in complex with the G protein CCD revealed that these mAbs bind two distinct G protein epitopes [40]. While both mAbs bind to RSV G with very high affinity (low picomolar), the 3D3 epitope is comprised of highly conserved residues whereas the 2D10 epitope is a mixture of conserved and non-conserved residues. Previous studies in mice examining RSV A2 and long strains support 3D3 as a prophylactic or therapeutic countermeasure to control the RSV challenge; however, there is no information regarding the evaluation of 2D10 [36,37]. In this study, we sought to determine if the different binding of these mAbs affected in vivo neutralization or the immune responses to RSV infection. 

The ability to neutralize RSV Line19F was examined in prophylactic and therapeutic regimens of mAbs 3D3, 2D10, or palivizumab treatment compared to isotype control treatment mAb treatment. As expected, prophylactic treatment with the anti-RSV mAbs administered 24 hpi neutralized RSV Line 19F. At 72 hpi, mice treated with mAbs 3D3 or 2D10 significantly neutralized RSV Line19F compared to isotype control mAb-treated mice. This demonstrated for the first time that 2D10 is capable of in vivo neutralization and is especially notable as mAb 2D10 lacks neutralization capacity in vitro in immortalized cells even with the addition of complement [39]. Thus, anti-G protein mAbs are capable of neutralizing RSV when delivered prophylactically or therapeutically in mice.

RSV Line19F is a chimeric RSV A2 virus strain that induces a G protein-associated Th2-type biased immune response including increased IL-13, and causes distinct lung pathology in mice [44,80,81]. Mice are semi-permissive to RSV and thus their translational ability is reduced due to the lack of disease. However, Line19F improves the translational assessment of therapeutic mAbs and may aid RSV vaccine development [82]. In this prophylactic model, treatment of mice with mAb 3D3 significantly (*p* < 0.05) reduced total BAL leukocytes compared to isotype control, although mAb 2D10 and palivizumab substantially reduced BAL cell numbers as well. Notably, only prophylactic treatment with mAb 3D3 was able to significantly (*p* < 0.05) reduce BAL eosinophils which is important as pulmonary eosinophilia may be associated with enhanced respiratory disease. Taken together, these data indicate that prophylactic mAb 3D3 treatment can modify the BAL cell influx during RSV infection; however, the differences are subtle compared to mAb 2D10 or palivizumab treatment. 

Treatment of mice at 72 hpi with mAbs 3D3 and 2D10 did not affect BAL leukocytes compared to isotype control mAb treatment. Intriguingly, palivizumab treatment substantially increased BAL leukocytes compared to anti-G protein mAbs and isotype control mAb treatments. Interrogation of BAL leukocytes revealed that palivizumab treatment at 72 hpi substantially increased many cell types including T cells, NK cells, and eosinophils. This increase in BAL cells following palivizumab treatment was unexpected as previous studies evaluating palivizumab did not show increased BAL cells in treatment models [35,36]. One major difference between those studies and the present study is the low dose of mAb utilized here (i.e., 1 mg/kg) compared to those studies that used higher doses of palivizumab (e.g., 5–15 mg/kg) [35,36]. It is possible the inadequate RSV neutralization resulted in higher BAL cell influx. It is known that suboptimal doses of palivizumab may result in antibody-dependent enhancement (ADE) [83], and in the cotton rat model, enhanced BAL influx has been shown to occur when animals were vaccinated with low doses of F protein inducing suboptimal anti-F protein immune responses [84]. Additionally, it has been shown that palivizumab treatment may increase BAL cells and inflammatory and mucogenic cytokines after re-infection in a neonate mouse model [56]. However, the precise mechanism of the BAL cell increase remains unknown. 

Cytokine analysis showed that anti-G protein and anti-F protein prophylactic mAb treatment decreased IFNγ, IL-4, and IL-13 (*p* < 0.05), and RANTES (*p* > 0.05). Interestingly, mAb 2D10 and palivizumab prophylaxis significantly (*p* < 0.05) reduced IL-4 concentrations (~50% reduction) in the lung while mAb 3D3 treatment reduced concentrations by ~25%. Therapeutic treatment with anti-G protein or anti-F protein mAbs did not significantly decrease IFNγ or IL-4, but mAb 3D3 treatment significantly (*p* < 0.05) reduced IL-13 (~50% reduction) which is correlated with mucogenic disease in RSV Line19F infected mice. It is unknown how mAb 3D3 reduced IL-13 without a significant reduction in IL-4 in this model as both are canonical Th2-type cytokines, but IL-4 and IL-13 may be regulated differently in non-canonical pathways [85]. Palivizumab treatment resulted in nearly a 3-fold increase in RANTES. The function of RANTES is in leukocyte recruitment including T cells, NK cells, and eosinophils [86,87,88], and increased RANTES expression may have contributed to the increase in pulmonary leukocytes cells observed in palivizumab-treated mice. 

## 5. Conclusions

The results of this study highlight some of the subtle variances in prophylactic or therapeutic treatment with anti-F or anti-G protein mAbs, and suggest that future evaluation of mAb 3D3 as a post-exposure therapeutic option should be considered as this broadly neutralizing human mAb is able to block CX3C-CX3CR1 chemotaxis, improve early antiviral IFN responses, neutralize RSV, and reduce IL-13. Future studies should also consider testing combinations of γ1 and γ2 binding anti-G mAbs, and anti-F and anti-G mAb cocktails to further improve the correlation between RSV disease in the mouse and cotton rat models. 

## Figures and Tables

**Figure 1 viruses-15-01067-f001:**
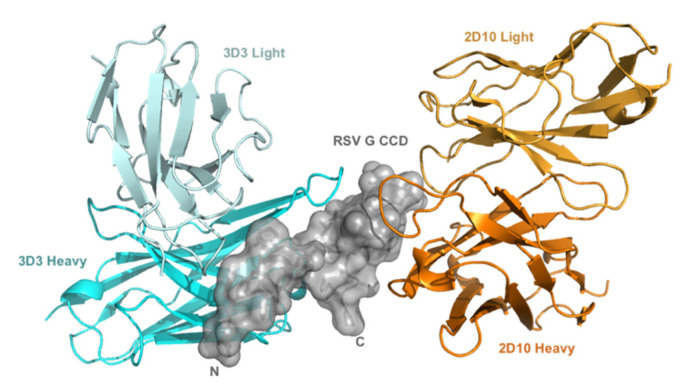
mAbs 3D3 and 2D10 bind distinct conformational epitopes. Distinct binding of 3D3 (cyan) and 2D10 (orange) to RSV G protein central conserved domain (CCD) (gray). Variable domains of mAb 2D10 (orange; PDB code 5WN9) and mAb 3D3 (cyan; PDB code 5WNA), when bound to overlaid RSV G CCD structures are displayed.

**Figure 2 viruses-15-01067-f002:**
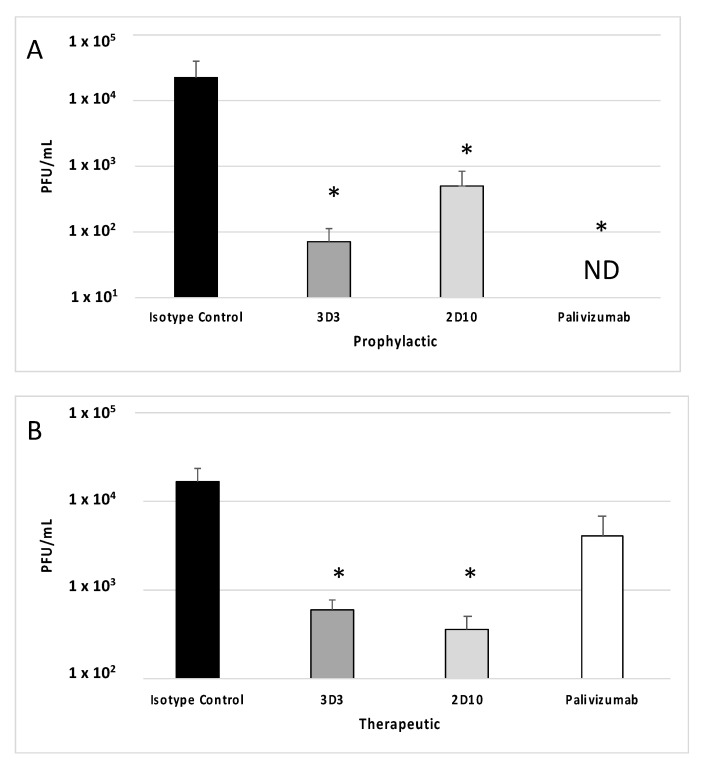
Therapeutic anti-G protein mAb treatment neutralizes RSV. Mice were (**A**) prophylactically (24 h prior to challenge) or (**B**) therapeutically (72 h post challenge) treated with 1 mg/kg isotype control, 3D3, 2D10, or palivizumab. Mice were challenged with 10^6^ PFU Line19F, and lungs were harvested on day 5 pi. Plaque assays were performed, and plaques were enumerated from n = 5 mice/group. Graphs represent the mean +/− SEM of PFU/mL lung homogenate. ND = Not detectable. * *p* < 0.05 compared to isotype control by one-way ANOVA with Bonferroni’s Correction (n = 5 mice/group).

**Figure 3 viruses-15-01067-f003:**
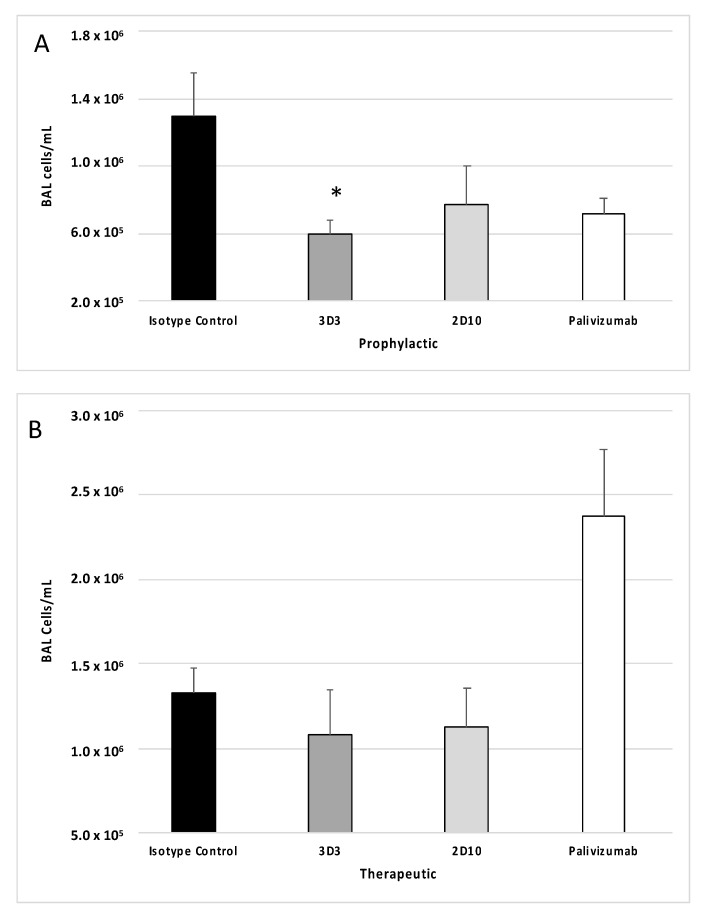
Anti-G and anti-F protein mAbs affect bronchioalveolar lavage (BAL) cell leukocyte recruitment following RSV Line19F infection. Mice were (**A**) prophylactically (24 h prior to challenge) or (**B**) therapeutically (72 h post challenge) treated with 1 mg/kg with isotype control mAbs, or mAbs 3D3, 2D10, or palivizumab. Mice were challenged with 10^6^ PFU Line19F, and BAL cells were collected on day 5 pi. Bars represent the mean ± SEM BAL cells. * *p* < 0.05 compared to isotype control as determined by one-way ANOVA with Bonferroni’s Correction (n = 5 mice/group).

**Figure 4 viruses-15-01067-f004:**
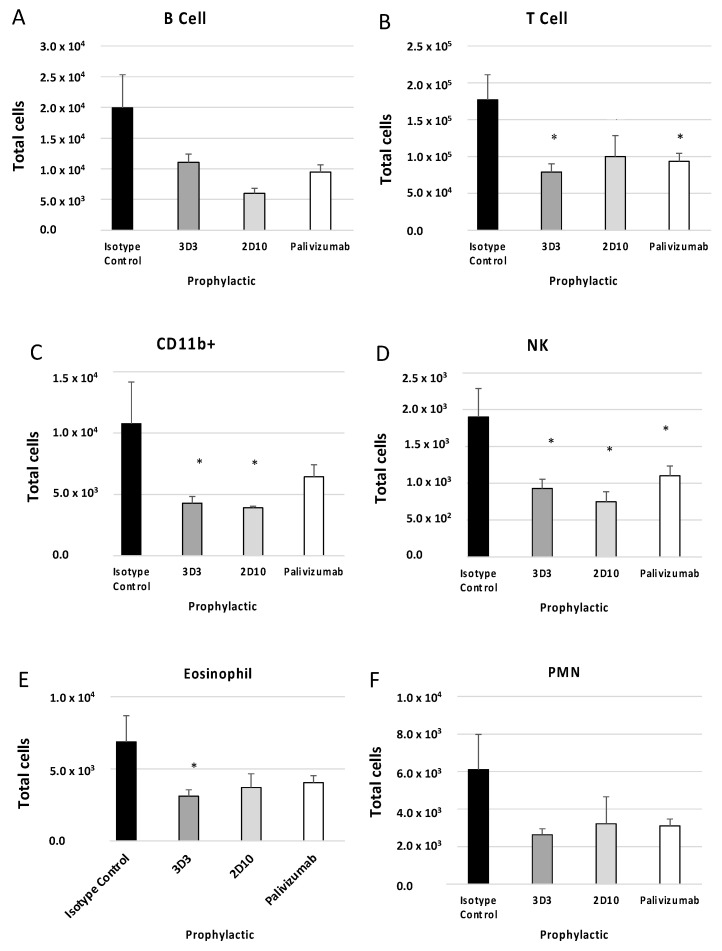
Prophylactic treatment with anti-G or anti-F protein mAbs affects BAL types. Mice were prophylactically (24 h prior to challenge) treated with 1 mg/kg with isotype control, 3D3, 2D10, or palivizumab. Mice were challenged with 10^6^ PFU Line19F, and BAL cells were collected on day 5 pi. (**A**) B cells = SSC^lo^ B220^+^, (**B**) T cells = SSC^lo^ CD3^+^, (**C**) CD11b^+^ (monocytes, macrophages, dendritic cells) = CD3^−^ CD11b^+^, (**D**) eosinophils = Ly6G^−^ CD11b^+^ SiglecF^+^, (**E**) NK cells = CD3^−^ DX5^+^, (**F**) PMNs (neutrophils) = Ly6G^+^ CD11b^+^ SiglecF^−^. Bars represent the mean ± SEM BAL cells. * *p* < 0.05 compared to isotype control as determined by one-way ANOVA with Bonferroni’s Correction (n = 5 mice/group).

**Figure 5 viruses-15-01067-f005:**
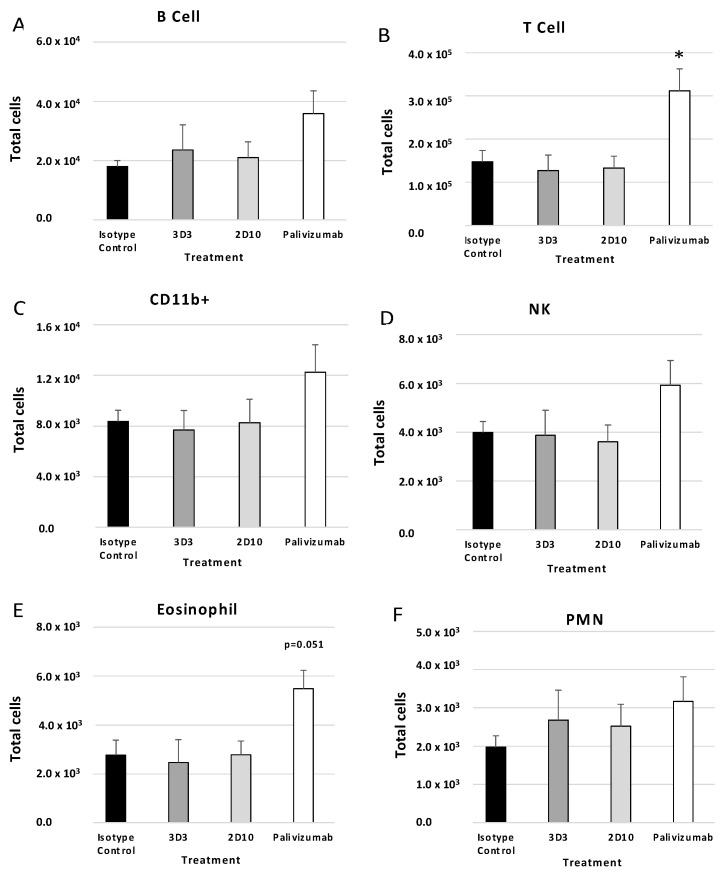
Therapeutic treatment with anti-G or anti-F protein mAbs affects BAL types. Mice were therapeutically (72 h post-challenge) treated with 1 mg/kg with isotype control, 3D3, 2D10, or palivizumab. Mice were challenged with 10^6^ PFU Line19F and BAL cells were collected on day 5 pi. (**A**) B cells = SSC^lo^ B220^+^, (**B**) T cells = SSC^lo^ CD3^+^, (**C**) CD11b+ (monocytes, macrophages, dendritic cells) = CD3^−^ CD11b^+^, (**D**) eosinophils = Ly6G^−^ CD11b^+^ SiglecF^+^, (**E**) NK cells = CD3^−^ DX5^+^, (**F**) PMNs (neutrophils) = Ly6G^+^ CD11b^+^ SiglecF^−^. Bars represent the mean ± SEM BAL cells. * *p* < 0.05 compared to isotype control as determined by one-way ANOVA with Bonferroni’s Correction (n = 5 mice/group).

**Figure 6 viruses-15-01067-f006:**
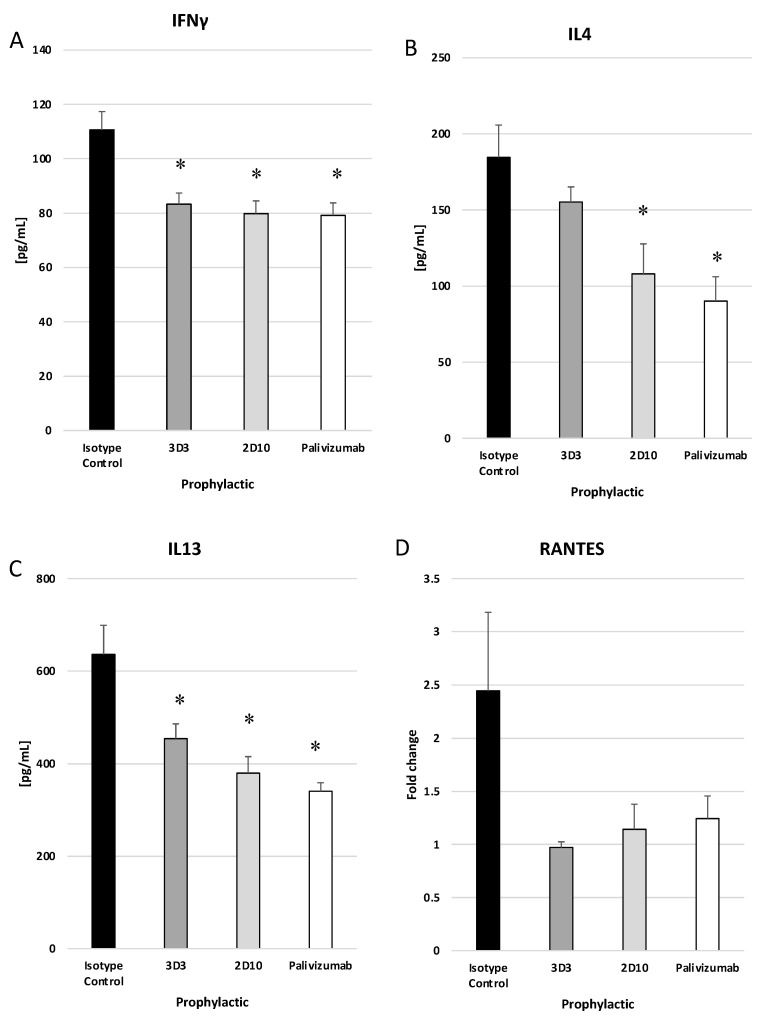
Prophylactic treatment with anti-G or anti-F protein mAbs reduce cytokine and chemokine responses. Mice were prophylactically (24 h prior to challenge) treated with 1 mg/kg with isotype control, 3D3, 2D10, or palivizumab. Mice were challenged with 10^6^ PFU Line19F, and lung homogenates were collected on day 5 pi. ELISAs were performed for (**A**) IFNγ, (**B**) IL4, and (**C**) IL13, and ΔΔCt PCR was performed to determine relative (**D**) RANTES transcript levels. Bars represent the mean ± ng/mL cytokine or fold change. * *p* < 0.05 compared to isotype control as determined by one-way ANOVA with Bonferroni’s Correction (n = 5 mice/group).

**Figure 7 viruses-15-01067-f007:**
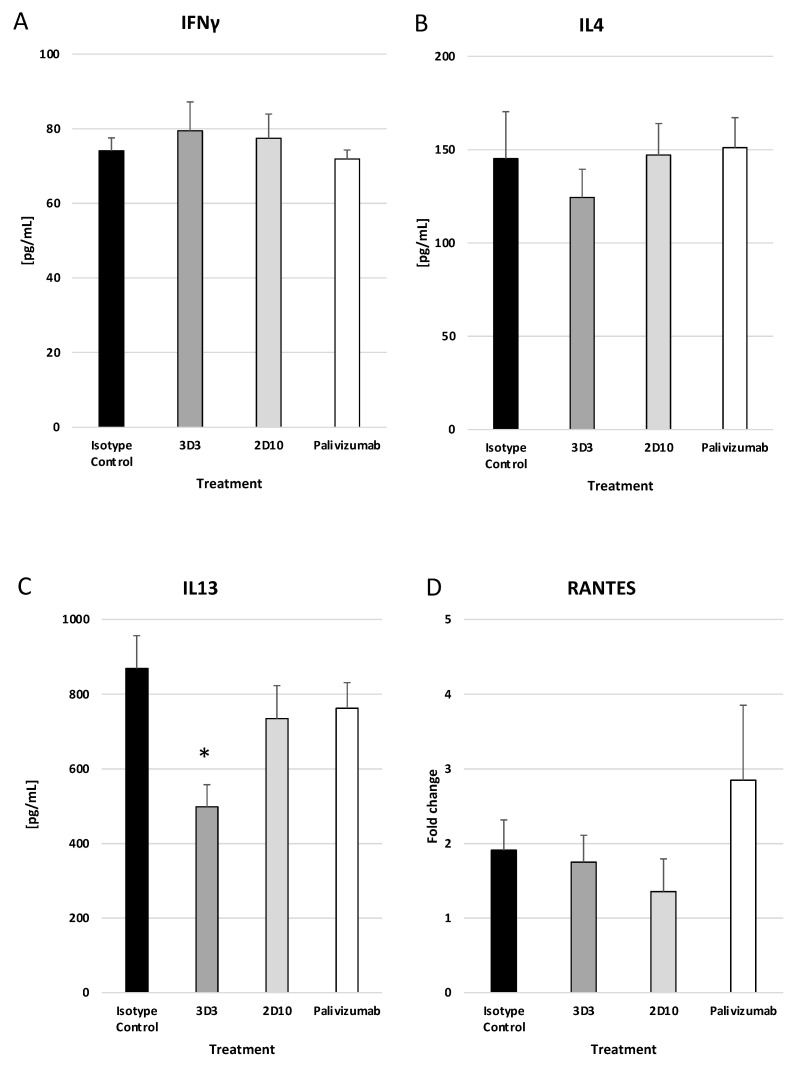
Therapeutic treatment with anti-G or anti-F protein mAbs does not greatly affect the cytokine response. Mice were therapeutically (72 h post challenge) treated with 1 mg/kg with isotype control, 3D3, 2D10, or palivizumab. Mice were challenged with 10^6^ PFU Line19F, and lung homogenates were collected on day 5 pi. ELISAs were performed for (**A**) IFNγ, (**B**) IL4, and (**C**) IL13, and ΔΔCt PCR was performed to determine relative (**D**) RANTES transcript levels. Bars represent the mean ± ng/mL cytokine or fold change. * *p* < 0.05 compared to isotype control as determined by one-way ANOVA with Bonferroni’s Correction (n = 5 mice/group).

## Data Availability

Data and materials are available upon reasonable request.

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
