# Peer review of "Immune Prophylaxis Targeting the Respiratory Syncytial Virus (RSV) G Protein"

_viruses, 2023, doi:10.3390/v15051067_

Round 1
Reviewer 1 Report
This paper focused on a monoclonal antibody (2D10) that reacts against as specific epitope of the respiratory syncytial virus (RSV) glycoprotein, G. The G protein is important for virion attachment whereby its CX3C motif binds to the cellular receptor, CX3CR. The authors performed a series of standard immunological analysis of interaction of this monoclonal, using FACS analysis, BALB/c mouse experiment to test for prophylactic and therapeutic protections against RSV infection, along with quantification of relevant immune cells (B cell, T cell, CD11b+, NK cells, eosinophils, PMN, etc) and cytokines, such as IFN gamma, IL4, IL13 and RANTES, a notable chemokine. Another monoclonal (3D3), against a different antigenic site of G was used for comparison, and interesting difference was seen between the two in terms of protection and disease severity. In some experiments, the authors also used the commercial monoclonal, Palivizumab, currently the only clinically viable RSV monoclonal (against F). It produced the expected results, serving as a form of "control". Lastly, attempts were made to correlate the G monoclonal results with available co-crystal structures. Overall, the paper is are very informative and of potential use in future studies.
I have a few comments of minor weaknesses, as itemized below.
1. The authors have used the F-mutant RSV, designated Line 19F, mainly to achieve a more robust infection in mice, since BALB/c is not permissive for net RSV replication. Nonetheless, RSV causes immunopathology when used in high amounts. One wonders whether use of wild type RSV, or a more permissive host (such as cotton rats) would have made a difference in the results of these studies. Some comments on this would be welcome. Perhaps cotton rat immune reagents are not available?
2. Besides offering structural confirmation that the monoclonals indeed bind to two specific sites (Gamma-1 and -2), the crystal structures seem to contribute nothing in this paper. They could be more relevant if used to explain some of the experimental data, such as prevention or relief of the RSV disease.
Reviewer 2 Report
Bergeron et al examine two human anti-G antibodies and compare their potential for prophylactic and therapeutic benefit against RSV in a mouse model. The anti-G antibodies are applied 24 h prior to challenge with line19 RSV or 72 hours post challenge, and compared to the FDA approved anti-F antibody Palivizumab. The impact on lung virus titer, BAL influx and immune cell types, and lung cytokines is examined. This is a fairly straightforward and well-written paper. Some of the data only confirm previous work with antibody 3D3, but inclusion of antibody 2D10, palivizumab, and therapeutic potential add important and new insights. The most significant findings are that anti-G antibodies reduce virus titer in vivo both prohylactically and therapeutically, whereas palivizumab is only effective prophylactically, and that both prohylactically-applied anti-G antibodies reduce lung influx of specific immune cell types such as T cells and eosinophils, as well as cytokines. All tested antibodies also reduce both Th1 and Th2 cytokines when applied prophylactically. This could be benificial though it is not clear whether that would result in an acceptable or improved Th1/Th2 balance. 3D3 also reduces IL-13 when applied at 72 h post challenge which could be a significant observation toward therapeutic potential. The therapeutic potential is especially noteworthy as a therapeutic drug is currently not available and, if effective, has potential for significant cost-savings compared to prophylactic drugs. Lastly, it is shown that palivizumab increases influx of several immune cell types when applied therapeutically. No plausible mechanism is offered, but the finding is discussed in light of other findings of enhanced disease upon suboptimal anti-F immunity. Combined with the finding that G is an IFN-antagonist, the authors make a good case that anti-G antibodies are important, and should be considered as, a post-exposure treatment. Only minor concerns were noted.
Minor concerns
Fig 4B. Palivizumab is lower and has a low SD, yet is not recognized as significant whereas 2D10, which has a larger SD, is.
Since anti-F and anti-G antibodies both have therapeutic potential, have the authors considered cocktails of anti-F and anti-G antibodies?
